# The Global Trend of Microplastic Research in Freshwater Ecosystems

**DOI:** 10.3390/toxics11060539

**Published:** 2023-06-17

**Authors:** Yaochun Wang, Guohao Liu, Yixia Wang, Hongli Mu, Xiaoli Shi, Chao Wang, Naicheng Wu

**Affiliations:** 1Department of Geography and Spatial Information Techniques, Ningbo University, Ningbo 315211, China; 2Pearl River Fisheries Research Institute, Chinese Academy of Fishery Sciences, Guangzhou 510380, China

**Keywords:** microplastics, freshwater environment, ecological impact, pollution, research progress

## Abstract

The study of microplastics and their impact on aquatic ecosystems has received increasing attention in recent years. Drawing from an analysis of 814 papers related to microplastics published between 2013 and 2022 in the Web of Science Core Repository, this paper explores trends, focal points, and national collaborations in freshwater microplastics research, providing valuable insights for future studies. The findings reveal three distinct stages of microplastics: nascent development (2013–2015), slow rise (2016–2018), and rapid development (2019–2022). Over time, the focus of research has shifted from “surface”, “effect”, “microplastic pollution”, and “tributary” to “toxicity”, “species”, “organism”, “threat”, “risk”, and “ingestion”. While international cooperation has become more prevalent, the extent of collaboration remains limited, mostly concentrated among English-speaking countries or English and Spanish/Portuguese-speaking countries. Future research directions should encompass the bi-directional relationship between microplastics and watershed ecosystems, incorporating chemical and toxicological approaches. Long-term monitoring efforts are crucial to assessing the sustained impacts of microplastics.

## 1. Introduction

The low decomposition rate and high productivity (300 million tons in 2020 [1]) of plastic have led to its increased presence in ecosystems, resulting in negative impacts on freshwater bodies [2]. Plastic pollution was listed as one of the “top 10 urgent environmental issues” by the United Nations Environment Assembly in 2014, and in 2018, World Environment Day was celebrated under the theme of “Plastic Wars Plastic Decisions” [3]. The occurrence of plastic in water has a toxic effect on organisms and poses significant ecological risks [4]. It is evident that, amidst the dramatic increase in demand for plastics, finding a balanced solution between environmental friendliness and plastic consumption is of utmost importance.

Plastic particles with a size smaller than 5 mm are commonly referred to as “microplastics” [5,6]. In the aquatic environment, plastic waste undergoes breakdown through mechanical overuse, photodegradation, oxidation, and weathering, resulting in the formation of small particles (<5 mm) known as secondary microplastics. On the other hand, plastic particles present in cosmetics and fibers in clothing are considered primary microplastics. Due to their small size, microplastics can be transported over long distances by wind, rivers, and ocean currents. Numerous studies have confirmed the impact of microplastics on organisms and ecosystems. Available information indicates that microplastic pollution is found in varying abundances across different locations, ranging from the equator [7,8] to the north and south Poles [9,10]. Previous studies have shown significant findings: Wright and Kelly found that the human excretory system eliminates 90% of microplastics through feces, while the remaining 10% is absorbed into the bloodstream [11]; the same study also revealed that nanoscale microplastics can pass through the brains of carp, leading to behavioral disorders. Additionally, Martins and Guilhermino discovered that microplastics can induce transgenerational effects in Daphnia, resulting in birth disorders and reduced growth rates in their offspring [12]. However, it is important to note that most toxicity studies have been conducted at high concentrations that are not environmentally relevant and often involve pristine polymers. In summary, global microplastic pollution has the potential to significantly impact ecosystems and human health. Therefore, ecological and environmental studies on microplastics are of great significance [13,14].

The sourcing, storage, pollution, and utilization of freshwater, a vital resource for irrigation and sustaining life on land, constitute a significant global concern [15,16,17,18,19]. Microplastics have been found in rivers and lakes [1] as well as wetlands [20]. Several highly cited review articles [1,20,21] have made valuable contributions by providing comprehensive insights into the methodologies, characteristics, distribution, and pollution associated with microplastics. However, these reviews primarily rely on traditional approaches such as summarizing the existing literature, which may result in limited coverage of the available research. To address this limitation, an analysis of trends, focal points, and national collaborations in microplastic research specifically focused on freshwater environments can offer a macro-level summary of the current state of knowledge. This approach, based on a comprehensive examination of numerous publications, provides a valuable resource for future research endeavors in the field of microplastics in freshwater environments.

With the increasing number of publications, publishing institutions, and international collaborations, there is a growing need for an objective and macroscopic research methodology. Bibliometrics plays a crucial role in assessing and analyzing researchers’ productivity [22], inter-institutional collaborations [23], the impact of national research investments on research and development [24], and the quality of scholarly work [25]. By analyzing key information from a large number of publications and utilizing visual methods to represent the intersection, derivation, and interaction between research objects, bibliometrics can help predict trends, explore interrelationships, and uncover hotspots and frontiers in research [26]. However, few articles have combined bibliometrics and the literature reviews to explore the progress of microplastic research in freshwater ecosystems. Therefore, this paper employs bibliometric analysis to summarize research trends, focal points, and international collaborations in the field of freshwater microplastics. Based on the selected keywords (TI = (microplastic*) AND TI = (fresh water or freshwater or plain water or lake or river or canal or stream or wetland or wetland or glacier or ice or groundwater or groundwater or underground water)) and the defined selection criteria, articles published since 2013 are included in this study. The objectives of this research are as follows: (i) to identify research trends in publications on microplastics in freshwater ecosystems from the period 2013 to 2022; (ii) to identify the focal points of publications on microplastics in freshwater ecosystems from 2013 to 2022; and (iii) to explore the trends of international cooperation on microplastics in freshwater ecosystems from 2013 to 2022.

## 2. Materials and Methods

### 2.1. Data Sources

Based on the Web of Science Core Collection database (including SCI source journals, SSCI source journals, CCR source journals and CI source journals), publications were searched using the following search term: TI = (microplastic*) AND TI = (freshwater or plain water or lake or river or canal or stream or wetland or glacier or ice or groundwater or underground water) to filter the articles in January 2023. After filtering and eliminating irrelevant literature, 814 articles related to the topic were selected. Since the articles were mainly distributed after 2013, this article focuses on the articles from 2013 to 2022 for analysis. It should be noted that this paper only analyzes articles from the core database and therefore may not include all studies on microplastics in freshwater environments.

### 2.2. Research Methodology

In this paper, we analyze the research focal points on microplastics in freshwater ecosystems from different periods by combining bibliometrics and the literature reviews. We used R software [27,28] to analyze trends in publication and international cooperation. Additionally, we utilized VOSviewer software [29] to analyze focal points at different periods. Specifically, we identified high-frequency keywords (focal points) based on the number of occurrences in the article titles and considered collaboration between authors from different countries as international collaboration. Notably, in the analysis of focal points with VOSviewer software, we selected keywords with more than five occurrences in 2013–2017 to be displayed in the figure, and keywords with more than ten occurrences in 2018–2022 to be shown. This selection was necessary due to the large number of keywords, and we focused on displaying high-frequency words. Regarding the analysis of country cooperation, we chose to display collaborations with more than two occurrences in 2013–2017 in the figure, and collaborations with more than five occurrences in 2018–2022.

## 3. Results

### 3.1. Analysis of Publication Numbers

Based on the search results, we analyzed the publication trends of microplastics in freshwater ecosystems (Figure 1). Although the first article on microplastics was published in 1977 [30], there was limited attention paid to freshwater microplastics before 2013. However, our results demonstrate a steady increase in the number of published articles each year from 2013 to 2022. Based on the annual publication numbers, we divided the research development process into three stages. The first stage, from 2013 to 2015, can be characterized as the nascent development stage, with a small number of papers and a slow rate of increase. The average annual number of publications during this stage was below 5. The second stage, from 2016 to 2018, marked the slow rise stage, with an average of 25 publications per year, resulting in a total of 75 papers. Finally, the period from 2019 to 2022 represented the rapid development stage, with a total of 725 articles published, accounting for 90% of the total literature. The annual rate of increase during this stage exceeded 50%. Overall, there has been a consistent and gradual increase in the number of publications related to microplastics in freshwater ecosystems.

### 3.2. Research Trends

Figure 2 illustrates the most frequently occurring keywords in the titles of the 814 articles and their associations with other keywords. Figure 2a provides an overview of the overall research focal point and trends from 2013 to 2022. Additionally, Figure 2b presents the focal points specifically from 2013 to 2017, while Figure 2c focuses on the period between 2018 and 2022. In the earlier years (2013–2017), the research focal point, as shown in Figure 2b, exhibited a certain level of repetition. The main areas of focus were “polyethylene”, “surface”, “effect”, “microplastic pollution”, and “tributary”. These 21 keywords can be categorized into three topics: sources of microplastics (green), distribution patterns and methods (blue), and environmental impacts (red).

However, as depicted in Figure 2c for the years 2018 to 2022, the research focus on microplastics expanded significantly. The emphasis shifted towards “toxicity”, “species”, “organism”, “threat”, “risk”, and “ingestion”. The 42 keywords in this period are grouped into two prominent topics: sources and distribution patterns (green) and environmental impacts (red). Notably, the proportion of environmental impacts increased compared to the earlier years. Overall, the research conducted between 2018 and 2022 exhibited a wider range of focal points and research areas than that of 2013–2017. The research focal point gradually shifted from the characteristics of microplastics to the environmental impacts they pose.

### 3.3. Trends in International Cooperation

The number of articles published by countries has shown an overall increase, as depicted in Figure 3 and Table 1 (and Appendix A). International cooperation in microplastics research has also witnessed strengthening, but there are still several countries that have not actively participated in such collaborations, including Sudan, Central Africa, Syria, Myanmar, Afghanistan, and Venezuela. The collaborations that have occurred primarily involve English-speaking countries or English and Spanish/Portuguese-speaking countries. Currently, China, the UK, the US, and Germany exert more influence in microplastic research. Notably, Czechia, France, and Slovenia have produced higher-quality articles, as indicated by their higher average number of citations, during the period of 2013–2017. Similarly, Singapore has shown a similar trend from 2018 to 2022.

Figure 3a shows the international cooperation in freshwater microplastics research from 2013–2022. Figure 3b and Appendix A highlight that international cooperation in microplastic-related research was minimal from 2013 to 2017. However, over the past five years, international collaboration has increased notably between the Americas and Europe, as depicted in Figure 3c and Appendix A. During the period of 2018–2022, there has been a significant rise in connections between China and the USA, Australia, and Canada. Notably, the US, UK, Australia, and Canada have exhibited greater international activity. Overall, research on freshwater microplastics is increasingly establishing connections across continents, with the most closely linked collaborations involving China and the USA, China and Australia, China and Canada, the USA and Canada, and the USA and the UK.

Table 1 lists the top ten countries with the most cited publications under freshwater microplastics research. From 2013–2017, the United States ranked first in citations (4053), while Czechia had a smaller number of publications but ranked tenth (350) in citations and first in average citations, above the United States. There is little difference in the ranking of the UK, China, Canada, and Germany regarding the number of articles published and citations. Overall, the US had a more substantial influence during this period.

From 2018 to 2022, China has a clear advantage in terms of the number of publications (748) and citations (10,571), with more than four times the number of publications than the second place (2161), but the average citation of publications is relatively low. Singapore has the highest average number of citations (822), a large difference from second place. During the same period, the US ranked second in terms of the number of articles published but third in terms of citations, indicating a decline in the overall influence of the US during this period.

## 4. Discussion

### 4.1. Sources and Distribution Patterns of Microplastics from 2013 to 2017

The inclusion of terms such as “polypropylene (PP)” and “polyethylene (PE)”, as well as “polymer type”, from 2013 to 2017 suggests that the distribution of microplastics by polymer type was a major focus during this time period. PP and PE have been identified as primary sources of microplastics in extensive research and have remained significant concerns throughout the history of microplastic studies. PE is widely used in various industrial applications, such as packaging films for industrial and food products, shopping bags, garbage bags, and wrap-around films. Additionally, PP is mainly used in automobiles, appliances, and single-use packaging, with an average of 20% of the plastic parts used in every car worldwide being made of it, making it the most common source of microplastics [31]. The combined analysis demonstrates a high level of concern for both microplastics and their prevalence in freshwater between 2013 and 2017.

The presence of terms such as “tributary” and “surface” indicates the exploration of microplastic distribution areas from 2013 to 2017. Scholars during this stage have focused on discussing tributaries, rivers [32], estuaries [33], lakes [34], and streams [35]. Based on our literature summary, many studies during this period (2013–2017) have analyzed the distribution of microplastics in terms of surface water microplastic pollution [36,37,38] in relation to their quantity, abundance, shape, and color [35]. However, the distribution of microplastics can vary in different environments [36], and it is important to enhance the study of microplastics in different habitats. Zbyszewski and Corcoran suggested that the number and shape of microplastics can indicate whether they have undergone fragmentation, such as being subjected to wave action, sedimentation, or oxidation [39]. During 2013–2017, the focus was on the morphological characteristics of microplastics and their presence in the water column, although most attention was given to surface waters. Microplastics of different materials have different densities, and therefore, they can be distributed throughout the water column, with low-density microplastics typically occupying the surface water environment while high-density microplastics are often found in sediments and benthic organisms [40]. However, physical or chemical breakdowns can alter the size and structure of microplastics, thereby changing their position in the water column as a secondary effect [40].

In summary, the focal point for sources and distribution patterns of microplastics in 2013–2017 is the microplastic distribution and physical characteristics (e.g., number, size, distribution), while studies of environmental impacts are lacking.

### 4.2. Environmental Impacts from 2013 to 2017

The study of the sources and distribution patterns of microplastics eventually led to a discussion of their impact on water bodies. The inclusion of critical words such as “influence”, “concern”, and “organism” demonstrates the scholarly focus on the effects of microplastics on freshwater. However, the focus on these topics during this stage was not comprehensive and primarily relied on studying their interaction with the environment and organisms based on particle size, material, and shape [35,41] and their sorption effects.

Microplastics in the aquatic environment can directly or indirectly affect the quality of the abiotic environment. Plastics can impact the scattering of light in the aquatic environment, thereby affecting chemical cycling. Different concentrations of microplastics have different effects on the environment [42]. For example, estuaries with high concentrations of microplastic pollutants have more significant pollution effects [42]. In addition to interactions with the abiotic environment, microplastics have broader impacts by indirectly or directly affecting biological communities [43]. Although this topic has attracted some attention since 2013, few scholars focused on it before 2018. Some studies have demonstrated the uptake of microplastics by crabs, amphibians, and invertebrates [44,45]. However, there is uncertainty regarding whether ingestion of uncontaminated microplastics will impact the health of organisms [46]. Research has primarily focused on whether organisms ingest microplastics, while there is still limited research on the effects after ingestion.

### 4.3. Sources and Distribution Patterns of Microplastics from 2018 to 2022

More attention has been given to the freshwater environment since 2018, likely due to a greater awareness of its direct connection to human health. Studies conducted in freshwater environments such as the Kelvin River in the UK, Poyang Lake in China, the Lagoon of Bizerte in Tunisia, the Flemish Rivers in Belgium, and Vembanad Lake in India have revealed that microplastic concentrations in the freshwater range from 0.01 to 3 g/L [47,48,49,50].

The sources of microplastics include “polyvinyl chloride (PVC)”, “polyester”, “pellet”, and “fiber”, with fibrous microplastics being of particular concern from 2018 to 2022. Numerous studies have found high rates of fibrous microplastics in laboratory simulations of laundry processes, as well as in microplastic emissions from wool textiles during home laundering [51,52,53,54]. Freshwater microplastic samples also exhibited a significant amount of fibrous microplastics, up to 59% [55]. Polyvinyl chloride (PVC), a highly carcinogenic material capable of adsorbing and accumulating triclosan and affecting zebrafish, has been identified as a source of microplastics in the aquatic environment [56], leading to increased interest among scholars. PVC and polyester, due to their high densities, tend to sink to the bottom of rivers, while PP and PP are floating microplastics due to their low densities.

The final distribution of microplastics in the water column is also influenced by their degree of weathering, sorption, and rate of aging [57]. Denser microplastics may cause more lasting damage to the environment. Various environmental factors can affect the rate and mode of microplastic decomposition [58], and physical breakdown can be slower in freshwater compared to marine environments [55]. Some studies have shown that certain lake environments may experience more significant weathering of microplastic particles due to enhanced UV penetration [38]. Additionally, research has started to focus on domestic wastewater. Studying microplastics in domestic wastewater provides insights into the quantities and characteristics of microplastics released through household discharge. Overall, microplastic enrichment is more prominent in areas with intense human activity, and research is beginning to shift toward freshwater and urban environments that are in closer proximity to humans.

### 4.4. Environmental Impacts from 2018 to 2022

The largest proportion of the environmental impact of microplastics occurred between 2018 and 2022, with a greater focus on topics such as “species”, “transport”, “risk”, “adsorption”, “ingestion”, “interaction”, “threat”, “toxicity”, and “fish” compared to the period of 2013–2017. Toxicological and adsorption effects of microplastics: research on the toxicology of microplastics has primarily concentrated on aquatic organisms, with limited investigation into their effects on humans. It has been discovered that the breakdown of untreated plastic waste in water into microplastics can result in more complex ecological threats as they migrate and adsorb substances [59]. Numerous scholars have also observed that microplastics combine with harmful environmental pollutants such as additives, pesticides, heavy metals, and pharmaceuticals, forming more complex mixed pollutants. Consequently, the complex contaminants associated with microplastics are linked to various human diseases, including obesity, endocrine disorders, cancer, and cardiovascular problems, indicating that microplastic pollution poses a toxicological threat to human health [60,61].

Overall, the environmental impact of microplastics can pose a more serious toxicological threat, depending on the combination of contaminants. While microplastics tend to absorb pollutants from the environment [62], their sorption capacity varies among different types and environments [63]. This variation can be attributed to different microplastics having rubbery domains, functional groups, and polarity. Research suggests that polyethylene (PE) has a more flexible structure compared to other materials, making it more prone to adsorb organic compounds [64,65]. Older microplastics are more likely to adsorb contaminants compared to newly introduced ones, and the polarity of the microplastic and the contaminant during adsorption affects the process. For example, the amide-based polar polymer polyamide (PA) exhibits a higher adsorption capacity for polar antibiotics [66], while polar polystyrene (PS) has a higher affinity for polar nitrobenzene, and non-polar perfluorooctane sulfonamide (FOSA) has a higher adsorption capacity for non-polar PE [67].

Additionally, environmental factors such as pH, salinity, and the concentration of contaminants in the surroundings influence the sorption process. Studies have demonstrated that increasing salinity and pH both enhance the sorption capacity of PE [68]. Interestingly, microplastics sampled closer to the pollution source generally exhibit higher sorption capacities than those collected farther away. In summary, the sorption behavior of microplastics is influenced by both environmental factors and their physical properties. However, the ingestion of such contaminated microplastics by aquatic organisms can result in more severe and widespread harm and may even pose a risk to humans through trophic transfer.

Microplastics have an impact on aquatic organisms and the food web. The inclusion of keywords such as “fish” among the topics indicates that more scholars are beginning to investigate the actual effects of microplastics on organisms. In recent years, there have been an increasing number of experiments simulating the ingestion of microplastics by freshwater organisms to study their presence and effects [43,69,70]. These studies have demonstrated that once aquatic organisms ingest microplastics, their digestive systems are the first to be affected. For instance, here’s a study discovered that microplastics accumulate in the digestive tract, causing intestinal blockage, pseudo-satiation, and reduced food capacity [71]. Studies on crayfish have shown that their feeding rate and body weight decrease with increased ingestion of microplastics [72], while fish experience intestinal damage, such as rupture of intestinal villi and splitting of intestinal epithelial cells, due to microplastic ingestion [73]. Persistent damage to the digestive system can lead to more severe consequences, including growth inhibition, damage to the reproductive system, and reduced mobility [46]. Excessive ingestion of microplastics can also cause other adverse effects, including oxidative stress, altered enzyme levels, and physical organ damage [72,73]. It is important to note that these effects are based on currently observable levels, and microplastics that go unnoticed may gradually break down into smaller particles within the organism, penetrating the cell membranes and participating in the body’s circulation.

As previously discussed, chemically contaminated microplastics will carry toxic substances into organisms and the food chain, damaging the entire food web [62,74]. As early as 2014, some scholars have found that microplastics moved along the food web from lower to higher trophic levels [75]. In 2016, in the well-known experiment by Batel, an artificial food chain including Artemia to zebrafish was set up and found that microplastics can eventually accumulate in zebrafish via Artemia [76]. Following this, some scholars have discovered microplastics in salt, a finding that brought microplastics and humans closer together [77]. Subsequently, Karami et al. found microplastics in commercial salt from different countries, including Australia, France, Iran, Japan, Malaysia, New Zealand, and Portugal [78]. In summary, microplastics threaten freshwater organisms and food webs, and more understanding is needed to predict the future impact of microplastics.

## 5. Conclusions and Future Research

The main conclusions drawn from the visual analysis of publications, focal points, and international cooperation on microplastics in aquatic ecosystems are as follows:
(1)Number of literature phases: Between 2013 and 2022, publications on microplastics went through three stages: budding development (2013–2015), slow rise (2016–2018), and rapid development (2019–2022). The number of publications increased over time.(2)Trends in focal points: The research focus shifted from the basic morphological characteristics, distribution, and fundamental impacts of microplastics (2013–2017) to the complex impacts of microplastics (2018–2022). In the earlier period (2013–2017), the effects of microplastics were primarily studied in terms of their shape and polymer type. In the later period (2018–2022), the focus expanded to include species, organisms, transport, toxicity, and other related factors.(3)Trends in international cooperation: International cooperation in microplastics research has increased over time, with strong representation from countries such as China, the USA, Australia, and Europe. Collaborations often occur between English-speaking countries or between English and Spanish/Portuguese-speaking countries, indicating that language barriers may limit broader international collaboration. There are still many countries and regions globally, particularly in Africa, the Middle East, and South America, that are not extensively involved in international cooperation on microplastics research.

Future research should include the following:
(1)Strengthening the cross-analysis of microplastics with chemistry and toxicology to study their sorption-pollution effects with different pollutants. Currently, research on the sorption contamination of microplastics is not extensive enough. While there is a focus on sorption contamination with antibiotics, heavy metals, and other substances, the contamination effects and toxicological impacts of microplastics combined with microorganisms, bacteria, pesticides, new pollutants, and other substances have been less studied. Therefore, future studies should expand the classification of the sorption-pollution effects of microplastics for different contaminants.(2)Strengthening the long-term monitoring of microplastics to explore the actual pollution process over extended periods. Most current studies on the effects of microplastics are conducted in laboratory settings, where the microplastics retain their original qualities, shapes, and materials [79]. However, microplastics undergo changes in the environment over time. Currently, there is a lack of long-term monitoring data in the field of microplastics research, which can hinder scholars’ understanding of the actual contamination process of microplastics. Therefore, future research should focus on strengthening long-term monitoring studies to investigate the evolving relationship between microplastics and the environment.

## Figures and Tables

**Figure 1 toxics-11-00539-f001:**
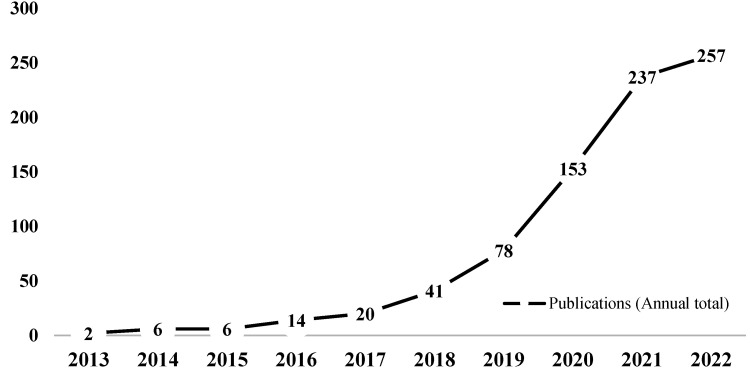
Publications in the field of microplastics in freshwater ecosystems from 2013 to 2022 (data from the Web of Science Core Collection). The horizontal axis represents the different years from 2013-2022 and the vertical axis represents the number of publications. The numbers on the line represent the number of publications corresponding to each year.

**Figure 2 toxics-11-00539-f002:**
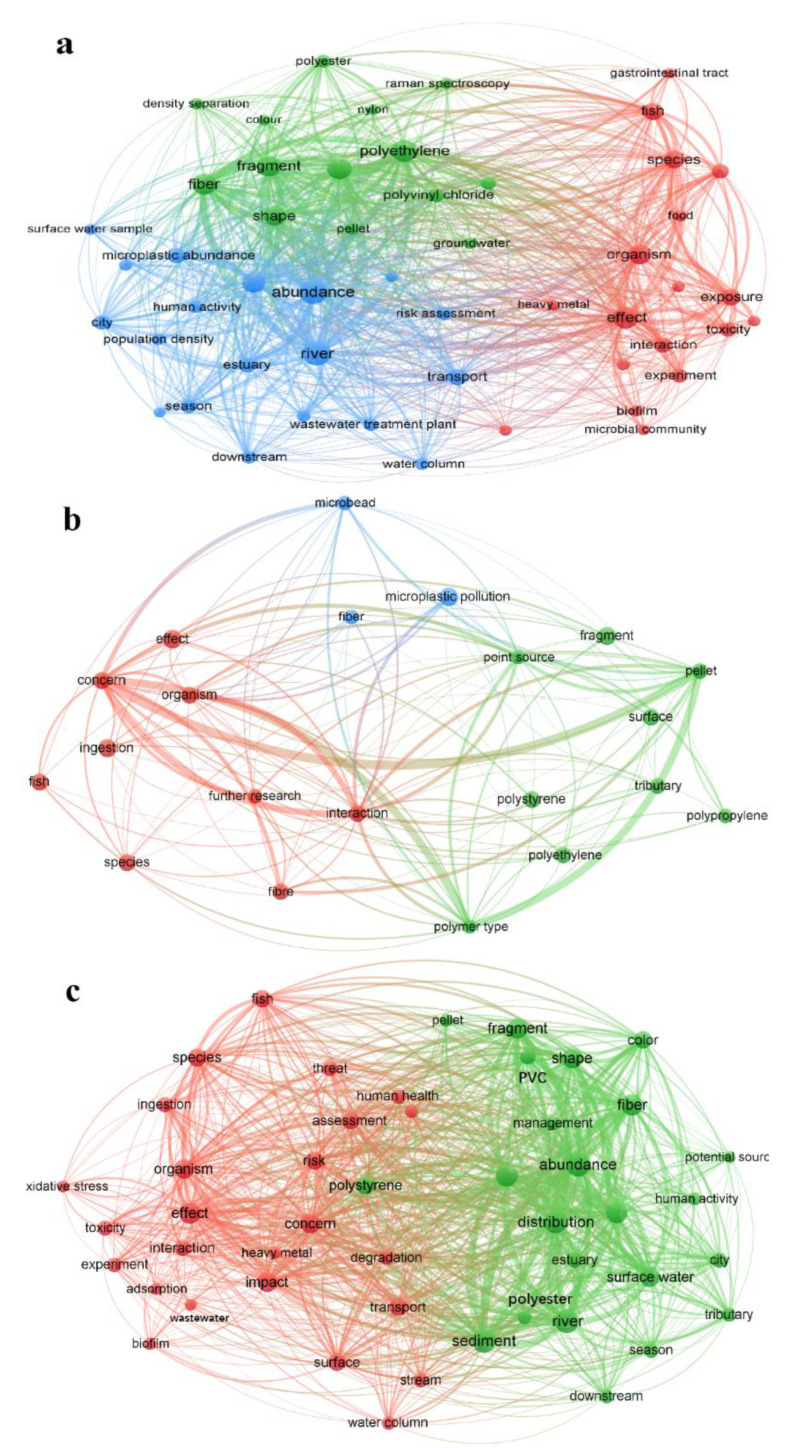
Research focal points and trends of microplastics in freshwater ecosystems over different periods. (**a**): 2013–2022; (**b**): 2013–2017; (**c**): 2018–2022. The size of the circles indicates the frequency of the keywords; the larger the circle, the higher the frequency of occurrence; the different colors symbolize the hot topics in which the keywords are found, with the keywords at the edges of the hot topics being transitional words; and the lines between the keywords indicate their connections to other keywords.

**Figure 3 toxics-11-00539-f003:**
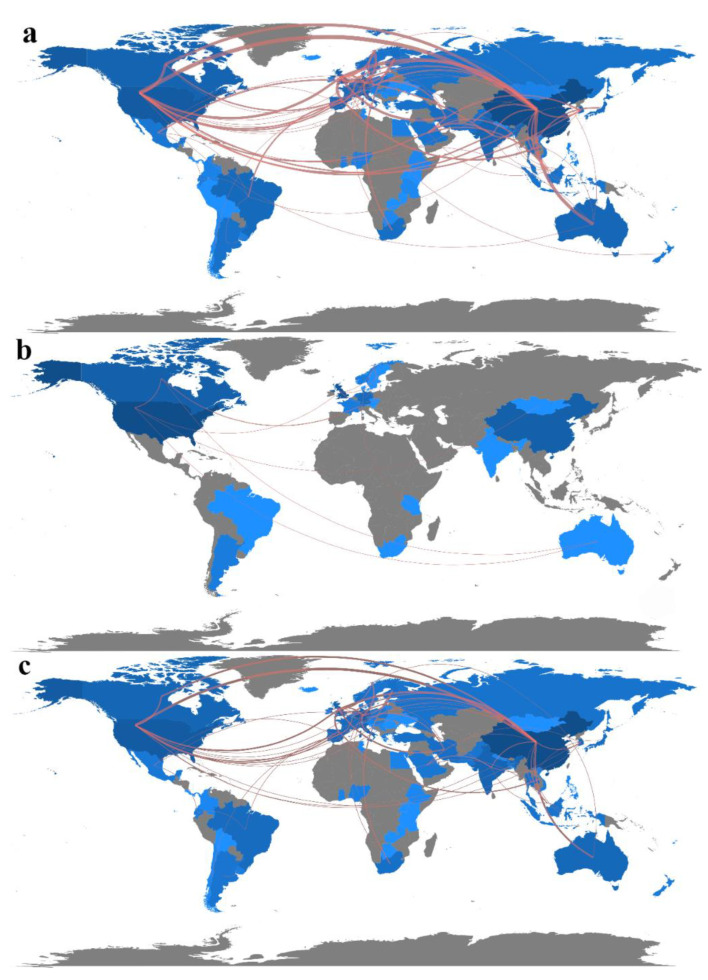
Temporal trend changes in international cooperation on microplastics in aquatic ecosystems. (**a**): 2013–2022; (**b**): 2013–2017; (**c**): 2018–2022. The shade of color indicates the number of articles published by countries; the darker the color, the more articles are published; red lines indicate international collaborations; the higher the volume of collaborative publications between two countries, the thicker the red line between them and the closer the relationship (We specify that the display starts after more than five international collaborations in 2013–2017 and after more than ten international collaborations in 2017–2022).

**Table 1 toxics-11-00539-t001:** Top 10 cited countries and publications from 2013 to 2022 (publications, number of citations—the total number of citations for articles published in this country).

2013–2017	2018–2022	2013–2022
United States (29, 4053)	China (748, 10,571)	China (883, 12,898)
United Kingdom (12, 3821)	United Kingdom (91, 2161)	United Kingdom (196, 6054)
China (12, 2314)	United States (160, 2001)	United States (122, 5983)
Germany (6, 1136)	Germany (143, 1957)	Germany (149, 3093)
Canada (17, 1057)	Italy (84, 953)	Canada (89, 1715)
Netherlands (6, 923)	Portugal (59, 912)	Netherlands (28, 1636)
Switzerland (4, 602)	Spain (63,827)	Italy (93, 1010)
France (3, 565)	Singapore (1, 822)	India (81, 1010)
Czechia (1, 350)	India (70, 763)	Portugal (67, 914)
Slovenia (3, 318)	Australia (52, 736)	France (52, 993)

## Data Availability

Not applicable.

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
