# Peer review of "The Global Trend of Microplastic Research in Freshwater Ecosystems"

_toxics, 2023, doi:10.3390/toxics11060539_

Round 1
Reviewer 1 Report (New Reviewer)
The paper titled with The Global Trend of Microplastic Research in Freshwater Ecosystems, overall it is an interesting review, however, there are some week points,
Abstract should be rewrite and could be very crisp.
Literature review is very shallow and most of the recent works are not cited properly.
For instance, Zhang et al. 2016 projected 250 MMT at 2025 during 2016 (7 years before), could authors can find any recent literature.
Plastic particles with a particle size of less than 5 mm are generally referred to as “microplastics” (Chen et al. 2019), why author used Chen 2019, since these size defination can be provided many authors quite before, for example Richard Thomson provided during 2004-2010.
Author why dont consider the microplastics in fiberous shape? for instance,
Since the most of the literature is not cited with microplastics from domestic washing
https://www.sciencedirect.com/science/article/pii/S0141391022000878
https://www.sciencedirect.com/science/article/pii/S0269749117341234?via%3Dihub
https://pubs.acs.org/doi/10.1021/acs.est.7b01750
English is fine however, there are some typo errors like the missing sections.
Author Response
The paper titled with The Global Trend of Microplastic Research in Freshwater Ecosystems, overall it is an interesting review, however, there are some week points,
Abstract should be rewrite and could be very crisp.
R1: Thank you very much for your suggestion, I have revised the summary and it is now much crisp.
Literature review is very shallow and most of the recent works are not cited properly.
For instance, Zhang et al. 2016 projected 250 MMT at 2025 during 2016 (7 years before), could authors can find any recent literature.
R2: Thank you very much for your suggestion, we have updated the references you mentioned as followes: “global plastics production reaches 300 million tonnes in 2020 (Padha et al. 2022)”.
Plastic particles with a particle size of less than 5 mm are generally referred to as “microplastics” (Chen et al. 2019), why author used Chen 2019, since these size defination can be provided many authors quite before, for example Richard Thomson provided during 2004-2010.
R3: Thank you very much for your suggestion, we have changed the reference article to a 2008 article by Moore and 2004 article by Thomson.
Author why dont consider the microplastics in fiberous shape? for instance,
R4: Thank you very much for your suggestion. At your suggestion, we have added a description related to fibrous microplastics in lines 273-276, we hope you are satisfied!
Since the most of the literature is not cited with microplastics from domestic washing
R5: Thank you very much for your suggestion, we've added some articles on domestic washing microplastics to line 273-276 of our manuscript, and we hope our revisions are to your satisfaction!

Reviewer 2 Report (New Reviewer)
The manuscripts presents a comprehensive review on the Global trends in MPs research and is focused on the freshwater MPs. The manuscript is well written and organized. Judging by the numerous corrections, the text has already been reviewed and revised.
There are only few comments:
Line 184: Exponential growth is observed only in the period of 2018-2021 in Fig. 1.
Figure 1. Please, add to the figure capture the source of the data (Web of Science Core collection).
Figure 2. Is it possible to size all three diagrams so that the entire figure fits on one page?
I would also recommend including Russian research in the analysis. May be use the review on MPs in Russian freshwater (Water, 2022, 14(23), doi: 10.3390/w14233909).
Author Response
The manuscripts presents a comprehensive review on the Global trends in MPs research and is focused on the freshwater MPs. The manuscript is well written and organized. Judging by the numerous corrections, the text has already been reviewed and revised.
There are only few comments:
Line 184: Exponential growth is observed only in the period of 2018-2021 in Fig. 1.
R1: Thank you very much for the suggestion, I have changed this to: “In general, there has been a gradual increase in the number of microplastics-related publications.”
Figure 1. Please, add to the figure capture the source of the data (Web of Science Core collection).
R2: Thank you very much for the suggestion, I have made changes in Figure 1 as follows: “Publications in the field of microplastics in freshwater ecosystems from 2013 to 2022 (data from Web of Science Core Collection).”
Figure 2. Is it possible to size all three diagrams so that the entire figure fits on one page?
R3: Thank you very much for the suggestion, based on your suggestions, we have made changes in lines 148-150 and hope you are satisfied!
I would also recommend including Russian research in the analysis. May be use the review on MPs in Russian freshwater (Water, 2022, 14(23), doi: 10.3390/w14233909).
R4: Thank you very much for the suggestion, we have added this article to line 222 in the text.

Reviewer 3 Report (New Reviewer)
Review: The Global Trend of Microplastic Research in Freshwater Ecosystems
The manuscript presents the results of a bibliographic study, but also reports on the contents of the publications that were investigated, which would be more suitable for a review. In that sense it is a mixture and it is not quite clear what the intended goal is: highlight the history of microplastic research or highlight the possible consequences of microplastics? In any case, I have detailed my critique below. My overall conclusion is that it requires major revisions at several points in the text.
General:
The references in the text are rather sloppy. I have indicated several occurrences, but I see in the list of references some 25 publications that are not referenced in the text and therefore should be removed, as well as references where the given name is used as if it were the family name – for instance no. 40: “Gert, Everaert, Lisbeth et al.” Both “Gert” and “Lisbeth” are given names. Reference 52 start with; Julie, C. Clearly “Julie” is the given name and the “C” hides the family name.
Abstract:
Lines 29-31: If we agree that microplastics do not belong in the environment (the portent of point 5), why should we continue to investigate the various aspects (points 1 to 4)?
Introduction:
Line 56 “Mattsson et al. (2017)” – this reference is missing.
Line 85: “D’Avignon, Gregory-Eaves et al.” – this is not in accordance with conventions. If an article has more than two authors, then only the name of the first author is given, followed by “et al.”. Please correct. There are more such occurrences (line 88 for example, Li, Busquet et al.).
Line 88: “Jamesd et al. (2020)” – this reference is missing.
Line 92: ”they still use a traditional approach” – what is the traditional approach and how does it differ from this study?
Line 126: “a large number of literature” -> “a large number of publications”
Lines 132-133: “While, …” – incomplete sentence. Please correct.
Line 137: “the period 2013 to 2022” – this period was selected on what grounds? The start of microplastics research? It seems rather arbitrary in the current presentation.
Materials and Methods:
Line 146: “this paper was searched” -> “papers were searched”
Lines 156-157: Is a period of ten years sufficiently long to be able to distinguish periods?
Line 158: “Null et al.” – this is a malformed reference. Please correct!
Line 162: “national” -> “international”
Lines 162-166: The formulation is rather odd. Why set keywords? How were these periods chosen? There is little information on how this was done. As it is presented now, it seems more a priori than determined from evidence.
Results:
Line 171: “began” -> “was published”
Lines 174-184: The numbers in the text do not correspond to numbers in figure 1.
Figure 1: What is the difference between the two subfigures? In the lower half the vertical axle is missing. The legend is incomplete.
Line 190: “most occurring” -> “most frequently occurring”
Section 3.2: Figure 2A is not referred to, 2B and 2C are.
Figure 2: There are six parts – A, B, C and a, b, c. The latter are not referred to in the legend. Please correct!
Table 2: What citations are meant here? Are they citations within the articles or are they the number of times the articles themselves have been cited? If the latter, why are these citing articles not included? (They cannot have been included, given the total number of articles.)
Table 2: “Czech” -> “Czechia”, “Slovebia” -> “Slovenia”
Line 306: “Gert et al. 2018” and “Everaert et al. 2018” are the same publication. Please correct.
Line 319: “Wagner et al. 2014” -> “Wagner and Scherer 2014” – the article has two authors only, another convention.
Line 332: “Zbyszewski et al. 2011” -> “Zbyszewski and Corcoran 2011”
Line 342: “density-cycled” – what is this? Please correct.
Line 358: “Ashton et al. 2005” – this reference is missing.
Line 373: “Christiane et al. 2011” – this looks to be a given name, not a family name. Please check.
Line 397: “In addition” – in addition to what?
Line 408: “Sruthy et al. 2017” – malformed reference: “S Sruthy, E V, et al. (2017)”. Please correct.
Line 414: “(Free et al. 2014)” – remove this, it has been stated at the start of the sentence.
Line 418: “Browne et al. (2015)” -> “Browne (2015)”
Line 418: “sewage treatment plants are a direct source” – treatment plants merely convey the sewage water to the receiving surface water system. They do not themselves produce microplastics, unless as part of their operation, but most come from the producers of the sewage in the first place.
Lines 431-433: That enrichment with microplastics occurs in the vicinity of human activities is not very surprising: we produce them, nature itself does not.
Lines 465-467: The absorption of contaminants by microplastics may occur, but it says very little about the amounts.
Lines 467 and 471: “Fred-Ahmadu et al 2019” and “Fred et al. 2019” are the same reference. Please correct!
Lines 471-474: Is this relevant? I suggest to remove it.
Line 481: Two articles “Wang et al. 2017” – rename to “2017a” and “2017b”.
Line 485: “Wang et al. 2015” – this reference has only two authors. See above.
Line 486: “FOSA” – what kind of plastic is that?
Lines 487-489: Incomplete sentence. Please correct.
Lines 490-492: “sorption … is usually higher than at sites far from the source …” – what is actually meant here? The sorption process or the amount of substance. The latter could be due to the concentrations of contaminants being higher near sources of contamination, but I do not see how the distance can affect the sorption process itself. Please explain!
Lines 496-499: Is this relevant? I suggest to remove it.
Lines 500-501: “the presence of organisms such as fish among the keywords” – please correct this, it is improbable that organisms and keywords physically interact, as suggested in this formulation.
Line 510: “Carlos et al. (2018)” – missing reference.
Lines 537-538: Rather alarmistic and we have no idea from the text what the concentrations are in the environment or at what levels this would indeed occur.
Line 540: “from lower to higher trophic levels” – hardly surprising, as the motion of food (and anything that comes with it) in the foodweb is by definition from lower to higher trophic levels.
Conclusions:
General comment: it is unusual to introduce such a large number of new references in the conclusions.
Lines 592-595: The articles that were investigated were explicitly selected for their focus on freshwater. Is it surprising that there is much less attention to the effects on humans?
Line 608: “hygromycin” – why is that isolated? It is an antibiotic too? It could be combined with the article by Li et al. 2018.
Lines 638-642: The subject of national laws is first mentioned here and without any references. That is highly unusual as well. Either remove it or provide relevant references to these laws and integrate it with the rest of the article.
Lines 643-657: While I agree with the conclusion that we should stop producing plastics, the rest of the article provides little to no arguments to support this conclusion. And it is again a new subject.
References:
The list of references needs to be thoroughly corrected, as there are many mistakes and omissions, as well as some 25 references that do not occur in the text.

I have indicated a number of locations in the text where wrong words are used or the sentences are incomplete. There are more such occurrences, though overall the quality is sufficient.
Author Response
Review: The Global Trend of Microplastic Research in Freshwater Ecosystems
The manuscript presents the results of a bibliographic study, but also reports on the contents of the publications that were investigated, which would be more suitable for a review. In that sense it is a mixture and it is not quite clear what the intended goal is: highlight the history of microplastic research or highlight the possible consequences of microplastics? In any case, I have detailed my critique below. My overall conclusion is that it requires major revisions at several points in the text.
General:
The references in the text are rather sloppy. I have indicated several occurrences, but I see in the list of references some 25 publications that are not referenced in the text and therefore should be removed, as well as references where the given name is used as if it were the family name – for instance no. 40: “Gert, Everaert, Lisbeth et al.” Both “Gert” and “Lisbeth” are given names. Reference 52 start with; Julie, C. Clearly “Julie” is the given name and the “C” hides the family name.
R: Thank you very much for your review of the article, we have made adjustments to the article based on your suggestions, especially in the references section. We hope that our changes will satisfy you!
Abstract:
Lines 29-31: If we agree that microplastics do not belong in the environment (the portent of point 5), why should we continue to investigate the various aspects (points 1 to 4)?
R1: Thank you very much for your suggestion, in response to your comments and those of other reviewers, we have changed this paragraph to read: “Future research should include: 1) to study the bi-directional relationship between microplastics and watershed ecosystems based on chemistry and toxicology; 2) to strengthen long-term monitoring of microplastics and assess the long-term impact of microplastics; 3) to formulate laws and regulations related to microplastics and solve the problem of microplastic pollution fundamentally”. Hope you will be satisfied with the changes I have made.
Introduction:
Line 56 “Mattsson et al. (2017)” – this reference is missing.
R2: Thank you very much for your suggestion, I have added this reference.
Line 85: “D’Avignon, Gregory-Eaves et al.” – this is not in accordance with conventions. If an article has more than two authors, then only the name of the first author is given, followed by “et al.”. Please correct. There are more such occurrences (line 88 for example, Li, Busquet et al.).
R3: Thank you very much for your suggestion, we have modified references like this.
Line 88: “Jamesd et al. (2020)” – this reference is missing.
R4: Thank you very much for your suggestion, I have added this reference.
Line 92: ”they still use a traditional approach” – what is the traditional approach and how does it differ from this study?
R5: Thank you very much for your suggestion, the traditional approach here refers to summarising articles through a literature review, which is very in-depth but does not cover a large number of relevant articles. This article combines a literature meta-visualisation with a literature review to analyse research hotspots in the context of a large number of articles. The main difference between this article and the traditional approach is therefore the different amount of article context used. In the article, we have added the relevant explanations as follows: “However, they still use a traditional approach (summary of articles through literature review) that can lead to problems due to low literature coverage and subjectivity.”.
Line 126: “a large number of literature” -> “a large number of publications”
R6: Thank you very much for your suggestion, we have already made changes on the line.
Lines 132-133: “While, …” – incomplete sentence. Please correct.
R7: Thank you very much for your suggestion, we have changed the word "While" to "However".
Line 137: “the period 2013 to 2022” – this period was selected on what grounds? The start of microplastics research? It seems rather arbitrary in the current presentation.
R8: Thank you very much for your suggestion, using our keywords and selection criteria, the earliest articles we extracted were from 2013, and 2023 is not currently closed, so we examined articles from 2013-2022. We have mentioned it in the methods section and in addition we have added the following explanation in the introduction: “According to the keywords we chose and the selection criteria, articles have been appearing since 2013” in line 87-88.
Materials and Methods:
Line 146: “this paper was searched” -> “papers were searched”
R9: Thank you very much for your suggestion, we have made the changes.
Lines 156-157: Is a period of ten years sufficiently long to be able to distinguish periods?
R10: Thank you very much for your suggestion, we have thought about this, but we feel that the increase in the number of articles from 2 in 2013 to 257 in 2022 represents a significant increase from 'nothing' to 'maturity'. It is true that there have been different phases during this period, so we would like to divide the development into different stages to express the research trends during the period 2013-2022. We've added this description to the introduction: “According to the keywords we chose and the selection criteria, articles have been appearing since 2013”.
Line 158: “Null et al.” – this is a malformed reference. Please correct!
R11: Thank you very much for your suggestion, we have made the changes as follows: “Null R, Team R, Null R, et al. (2011). R: a language and environment for statistical computing. Computing, 1, 12-21.”
Line 162: “national” -> “international”
R12: Thank you very much for your suggestion, we have made the changes.
Lines 162-166: The formulation is rather odd. Why set keywords? How were these periods chosen? There is little information on how this was done. As it is presented now, it seems more a priori than determined from evidence.
R13: Thank you very much for your attention. There may be a problem with my statement, the keywords mentioned here are not the ones used by WOS when looking for articles. What is meant here is that when visualising with VOSviewer software, we have chosen to show the higher frequency keywords as objects because there are so many of them. I have explained this in the manuscript. I hope you are satisfied with my answer.
Results:
Line 171: “began” -> “was published”
R14: Thank you very much for your suggestion, we have already made changes on the line.
Lines 174-184: The numbers in the text do not correspond to numbers in figure 1.
R15: Thank you very much for your suggestion, we've made the following changes: with the average annual number of publications below 5.
Figure 1: What is the difference between the two subfigures? In the lower half the vertical axle is missing. The legend is incomplete.
R16: Thank you very much for your attention to the article. The graph is intended to show the trend in the number of publications, but I don't quite understand what you mean by " subfigures ", " lower half " and the question. I am more than willing to make changes to the article, but if necessary I kindly ask you to explain your problem to me. Thank you so much!
Line 190: “most occurring” -> “most frequently occurring”
R17: Thank you very much for your suggestion, we have already made changes on the line.
Section 3.2: Figure 2A is not referred to, 2B and 2C are.
R18: Thank you very much for your suggestion, we've added relevant descriptions: fig. 2A shows the overall research focal point and trends from 2013 to 2022 in line 138-139.
Figure 2: There are six parts – A, B, C and a, b, c. The latter are not referred to in the legend. Please correct!
R19: Thank you very much for your suggestion, we changed all uppercase ABC to lowercase abc, we hope that our modifications will satisfy you
Table 2: What citations are meant here? Are they citations within the articles or are they the number of times the articles themselves have been cited? If the latter, why are these citing articles not included? (They cannot have been included, given the total number of articles.)
R20: Thank you very much for your suggestion, this refers to the total number of citations for articles published in this country. The number of citations therefore refers to the number of times the publication has been cited by other articles, not the articles themselves. I have explained in line 201 of the manuscript: “publications, number of citations-the total number of citations for articles published in this country”. I hope that my explanation is satisfactory to you.
Table 2: “Czech” -> “Czechia”, “Slovebia” -> “Slovenia”
R21: Thank you very much for your suggestion, on the line.
Line 306: “Gert et al. 2018” and “Everaert et al. 2018” are the same publication. Please correct.
R22: Thank you very much for your suggestion, we have modified it to: “making it the most common source of microplastics. (Gert et al. 2018)”.
Line 319: “Wagner et al. 2014” -> “Wagner and Scherer 2014” – the article has two authors only,
R23: Thank you very much for your suggestion, we have already made changes!
Line 332: “Zbyszewski et al. 2011” -> “Zbyszewski and Corcoran 2011”
R24: Thank you very much for your suggestion, we have already made changes!
Line 342: “density-cycled” – what is this? Please correct.
R25: Thank you very much for your suggestion, here we would like to talk about the water cycle due to different densities. We have explained this in the manuscript as follows: Microplastics can also become more dispersed in the water column if circulated by density cycled (water cycle due to different densities) or wind.
Line 358: “Ashton et al. 2005” – this reference is missing.
R26: Thank you very much for your suggestion, this reference has been added.
Line 373: “Christiane et al. 2011” – this looks to be a given name, not a family name. Please check.
R27: Thank you very much for your suggestion, we have already made changes to: Zarfl et al. 2011.
Line 397: “In addition” – in addition to what?
R28: Thank you very much for your suggestion, we deleted “In addition”.
Line 408: “Sruthy et al. 2017” – malformed reference: “S Sruthy, E V, et al. (2017)”. Please correct.
R29: Thank you very much for your suggestion, we have already made changes!
Line 414: “(Free et al. 2014)” – remove this, it has been stated at the start of the sentence.
R30: Thank you very much for your suggestion, we have removed it.
Line 418: “Browne et al. (2015)” -> “Browne (2015)”
R31: Thank you very much for your suggestion, we have already made changes!
Line 418: “sewage treatment plants are a direct source” – treatment plants merely convey the sewage water to the receiving surface water system. They do not themselves produce microplastics, unless as part of their operation, but most come from the producers of the sewage in the first place.
R32: Thank you very much for your suggestion, we have removed this section.
Lines 431-433: That enrichment with microplastics occurs in the vicinity of human activities is not very surprising: we produce them, nature itself does not.
R33: Thanks for the comment, it really is.
Lines 465-467: The absorption of contaminants by microplastics may occur, but it says very little about the amounts.
R34: Thanks for the comment, it really is.
Lines 467 and 471: “Fred-Ahmadu et al 2019” and “Fred et al. 2019” are the same reference. Please correct!
R35: Thank you very much for your suggestion, we changed Fred to Fred-Ahmadu.
Lines 471-474: Is this relevant? I suggest to remove it.
R36: Thank you very much for your suggestion, we have removed this sentence.
Line 481: Two articles “Wang et al. 2017” – rename to “2017a” and “2017b”.
R37: Thank you very much for your suggestion, I have made changes in lines 333 and 637-640.
Line 485: “Wang et al. 2015” – this reference has only two authors. See above.
R38: Thank you very much for your suggestion, I have revised the reference to read: (Wang and Shih 2015)
Line 486: “FOSA” – what kind of plastic is that?
R39: Thank you very much for your suggestion, here FOSA refers to perfluorooctane sulfonamide, as I have added an explanation in line 340 of the text.
Lines 487-489: Incomplete sentence. Please correct.
R40: Thank you very much for your suggestion, we already changed to: “In addition, environmental factors (e.g., pH, salinity, and the concentration of the contained contaminants) have an impact on the sorption process.”
Lines 490-492: “sorption … is usually higher than at sites far from the source …” – what is actually meant here? The sorption process or the amount of substance. The latter could be due to the concentrations of contaminants being higher near sources of contamination, but I do not see how the distance can affect the sorption process itself. Please explain!
R41: Thank you very much for your suggestion, here we wanted to say that more substances are adsorbed by microplastics in the source region, but I have removed this sentence due to the ambiguity of the expression. I hope you are satisfied.
Lines 496-499: Is this relevant? I suggest to remove it.
R42: Thank you very much for your suggestion, I have deleted this sentence.
Lines 500-501: “the presence of organisms such as fish among the keywords” – please correct this, it is improbable that organisms and keywords physically interact, as suggested in this formulation.
R43: Thank you very much for your suggestion, we have removed this sentence and hope you are satisfied.
Line 510: “Carlos et al. (2018)” – missing reference.
R44: Thank you very much for your suggestion, we have removed this reference.
Lines 537-538: Rather alarmistic and we have no idea from the text what the concentrations are in the environment or at what levels this would indeed occur.
R45: Thank you very much for your suggestion, we have deleted the latter part and replaced the sentence with the following: “In summary, microplastics threaten freshwater organisms and food webs, and more understanding is needed to predict the future impact of microplastics.”.
Line 540: “from lower to higher trophic levels” – hardly surprising, as the motion of food (and anything that comes with it) in the foodweb is by definition from lower to higher trophic levels.
R46: Thanks for the comment, it really is.
Conclusions:
General comment: it is unusual to introduce such a large number of new references in the conclusions.
R47: Thank you very much for your suggestion, we have removed references that were not needed here.
Lines 592-595: The articles that were investigated were explicitly selected for their focus on freshwater. Is it surprising that there is much less attention to the effects on humans?
R48: Thank you very much for your suggestion, I have removed this description.
Line 608: “hygromycin” – why is that isolated? It is an antibiotic too? It could be combined with the article by Li et al. 2018.
R49: Thank you very much for your suggestion, it is also an antibiotic, so I have removed that term.
Lines 638-642: The subject of national laws is first mentioned here and without any references. That is highly unusual as well. Either remove it or provide relevant references to these laws and integrate it with the rest of the article.
R50: Thank you very much for your suggestion, we have removed this description.
Lines 643-657: While I agree with the conclusion that we should stop producing plastics, the rest of the article provides little to no arguments to support this conclusion. And it is again a new subject.
R51: Thank you very much for your suggestion, we have removed this description.
References:
The list of references needs to be thoroughly corrected, as there are many mistakes and omissions, as well as some 25 references that do not occur in the text.
R52: Thank you very much for your review of the article, we have removed the irrelevant references

Reviewer 4 Report (New Reviewer)
In a peer-reviewed review, the authors presented an overview of the literature on the presence of microplastics in the aquatic environment. They pointed to the rapid development of interest in this topic, especially in the years 2019-2022. They showed the trends existing in individual years, from the interest in the presence of MP in water (type of polymer, particle characteristics) to the assessment of effects and risks to ecosystems.
When analyzing international cooperation, the authors of the paper should define what they mean by it: whether the nationality of the co-authors of the papers - which rather indicates the mobility of scientists, or joint research on water reservoirs belonging to different countries. The lack of such cooperation (line 570) does not necessarily mean a lack of interest in the subject, but more often due to the lack of funders or adequately equipped research institutions in many countries, e.g. Africa.
An important mistake that the authors should correct is the citation of papers published in years other than the discussed time period. For example, when discussing the period 2013-2017, they quote Ballent et al. 2012 (line 319) and Zbyszewski et al. 2011 (line329), and when discussing the period 2018-2022, they cite Svendsen et al. 2017, Browne et al. 2015 (line 418), Carr et al. 2016 (line 419).
Much of the literature concerns the presence of MP in seas or oceans or their effects on marine organisms (eg Ref. 3, 5, 7, 9, 10, 11...). This should not be the case for a freshwater survey. In addition, when discussing the effects on organisms, the results of both environmental and laboratory analyzes are quoted. Since, as the authors themselves write, laboratory analyzes are usually performed at concentrations much higher than those in the environment, these two types of research should be clearly separated and discussed.
Minor remarks:
line 380. polyester (PET). Not all polyester is PET.
lines 385, 389. Once the abbreviation PET has been introduced, it should be used consistently.
line 409. Are you sure that the concentration was so high?
line 427. ...to toothpaste?
Table 1. Should be Slovenia
line 479. ...electrodes?
line 499. Reference should be added.
line 592. The aim of the article is the MP research in freshwater ecosystems. Why was human impact research added as an indication for the future?
Author Response
In a peer-reviewed review, the authors presented an overview of the literature on the presence of microplastics in the aquatic environment. They pointed to the rapid development of interest in this topic, especially in the years 2019-2022. They showed the trends existing in individual years, from the interest in the presence of MP in water (type of polymer, particle characteristics) to the assessment of effects and risks to ecosystems.
When analyzing international cooperation, the authors of the paper should define what they mean by it: whether the nationality of the co-authors of the papers - which rather indicates the mobility of scientists, or joint research on water reservoirs belonging to different countries. The lack of such cooperation (line 570) does not necessarily mean a lack of interest in the subject, but more often due to the lack of funders or adequately equipped research institutions in many countries, e.g. Africa.
R: Based on your suggestion, we have removed this section of the manuscript (especially with regard to future research perspectives).
An important mistake that the authors should correct is the citation of papers published in years other than the discussed time period. For example, when discussing the period 2013-2017, they quote Ballent et al. 2012 (line 319) and Zbyszewski et al. 2011 (line329), and when discussing the period 2018-2022, they cite Svendsen et al. 2017, Browne et al. 2015 (line 418), Carr et al. 2016 (line 419).
R: Based on your suggestion, we have removed the articles by Ballent et al. 2012, Svendsen et al. 2017, Browne et al. 2015, Carr et al. 2016. In addition, we would like to keep the article by Zbyszewski: as this article is mainly about a phenomenon that, although not of this phase, sets the stage for this phase. Also we have cited other articles from this phase.
Much of the literature concerns the presence of MP in seas or oceans or their effects on marine organisms (eg Ref. 3, 5, 7, 9, 10, 11...). This should not be the case for a freshwater survey. In addition, when discussing the effects on organisms, the results of both environmental and laboratory analyzes are quoted. Since, as the authors themselves write, laboratory analyzes are usually performed at concentrations much higher than those in the environment, these two types of research should be clearly separated and discussed.
R: Thank you very much for your suggestion, at your suggestion, we have removed most of the articles about the marine. However, there are still some that I would like to cite and I hope you will allow them to exist (mainly in the introduction and explanation of the environmental behaviour of microplastics) e.g., Moret-Ferguson et al. 2010, Anthony and Andrady 2011, Arthur et al. 2009. This is mainly due to the common nature of the environmental behaviour of microplastics, but the lack of corresponding studies in freshwater. I hope you will allow this, but I am still willing to revise it if you find it inappropriate!
Minor remarks:
line 380. polyester (PET). Not all polyester is PET.
R1: Thank you very much for your suggestion, I have changed the polyester here to polyester as follows: the sources of microplastics include “polyvinyl chloride (PVC)”, “polyester”, “pellet”, “fiber”; On the other hand, PVC and polyester are denser than water.
lines 385, 389. Once the abbreviation PET has been introduced, it should be used consistently.
R2: Thank you very much for your suggestion, we have changed this to read: “On the other hand, PVC and polyester are denser than water.”
line 409. Are you sure that the concentration was so high?
R3: Thank you very much for your suggestion, yes, I am sure that this is the concentration, for details see ref: (Wenke et al. 2018), (Bordos et al. 2019), (Slootmaekers et al. 2018), (Sruthy and Ramasamy 2017).
line 427. ...to toothpaste?
R4: We have removed this part of the description.
Table 1. Should be Slovenia
R5: Thank you very much for your suggestion, I have modified it to “Slovenia”.
line 479. ...electrodes?
R6: Thank you very much for your suggestion, here I have modified it to “polarity”.
line 499. Reference should be added.
R7: Thank you very much for your suggestion, I have added references here.
line 592. The aim of the article is the MP research in freshwater ecosystems. Why was human impact research added as an indication for the future?
R8: Thank you very much for your suggestion, we have removed this suggestion and hope that our changes will satisfy you.

Round 2
Reviewer 1 Report (New Reviewer)
The manuscript has undergone significant enhancements as a result of the reviewer's feedback, and I highly endorse its publication in its current form.
Overall, English language has been improved, however, author should remove some typo errors.
Reviewer 4 Report (New Reviewer)
I do not have any additional comments.
This manuscript is a resubmission of an earlier submission. The following is a list of the peer review reports and author responses from that submission.
Round 1
Reviewer 1 Report
Authors provide an analysis of the evolution of 10 years of research on microplastics pollution in aquatic environments, underlying how the focus on sources, distribution and impacts have changed from 2016 to 2020 in respect to 2010 to 2015.
The manuscript is well structured, state of the art is well presented and discussed, research gaps on the topic are pointed out and the aspects which should be taken in consideration in the future and/or further investigated are suggested for better managment policies of plastic waste and for a greater understanding of impact of microplastics on aquatic ecosystems and human health.
From my perspective, the paper can be accepted without any further changes.
Reviewer 2 Report
This manuscript presents a limited review of publications from 2010 through January 2021 on the topic of microplastic in aquatic ecosystems. The authors present the number of publications from each country, with the resulting number of citations, and the distribution of keywords across two time periods (2010-2015, 2015-2020). There are some interesting ideas presented, but overall I am not convinced that the study adds to our overall understanding of trends in microplastic research in a meaningful way, especially given the very large number of review articles on the topic (>60 in 2022 alone). There are many errors and misstatements in the manuscript that suggest some carelessness in the presentation of the data and summarization of the studies at hand. As a result, even though some of the recommendations for research gaps are good, they don’t flow logically from the data as presented. Further, given the vast global interest in plastic pollution and the 18 month time period since the cutoff of this review, the field has moved on substantially and many of the points made are now moot. I fully understand the challenges in preparing a manuscript, but given that this paper hopes to be an up to date analysis, yet is 18 months behind the current trends in publishing, there are some issues with current relevance. Below I note some (but it is not an exhaustive summary) of the issues with the study in hopes that if the analysis is updated to be more recent, the writing and analysis are tightened up and better justified, the translation issues addressed and the errors fixed, the publication could be acceptable for a different submission.
Line 18 and in the discussion – the choice of these countries as “gaps” in research is questionable and doesn’t appear to emerge from the data at hand.
Line 41 – this implies that all plastic comes from microplastic that breaks down to microplastic. But much of the discussion later focuses on primary microplastics such as fibers and those in cosmetics. This should include both primary and secondary microplastics.
Line 53 - the impact on human health is largely unknown, so this sentence needs a qualifier – ie, “potential to seriously affect ecosystems and human health”
Lines 57+ - most studies of toxicity are at high concentrations that are not environmentally relevant and of pristine polymers. This is a shortcoming.
Line 91 – the cutoff of January 2021 leaves 18 months of publications. Based on the daily alerts that I receive from Google Scholar, at least 5 or more publications on this topic emerge each day, which would result in a near doubling of the literature in just the last 18 months. As such, a large swath of the current literature has been missed and the field has moved forward from what is noted here. There have been more than 60 review papers published in 2022 alone.
Line 109 – analyses ïƒ analyzes. The paper would benefit from editing by a native English speaker.
Table 1 – how were these studies selected? This appears to be a somewhat random selection of papers and there is not much analysis of what is presented. I think this table could be removed unless it is significantly augmented by trying to tally the number of studies on each of these topics. The summaries of the studies have a number of grammar errors and missing information that makes it so they are hard to understand – ie, The stomach contained 11%... 11% of what? Light and action per … per what? Reduces feeding and slowing down – the tense should be the same. How was this time period determined – this is 2013-2017, so crosses the two time periods of the study?
Line 169 – the colloborations listed are most often among English speaking countries, or English and Spanish/Portuguese speaking countries. This suggests that the language barrier is important and may be limited better international collaboration.
Line 172 – It is not clear how the determination that these countries are most closely linked was made. The methods could be clearer.
Figure 3 – I evaluated this in black and white, and I can’t tell the difference among the countries, and don’t understand why, for example, central Africa was called out. As noted above, the list of countries that are called out as lacking research do not clearly follow from the data as presented.
Table 2 – How was the location of authorship determined? Many papers are authored internationally, as described in the previous section of the paper.
Table 2 – over what period were the citations determined? Older papers are likely to have more citations than newer ones, so this analysis is a bit flawed because the values are averaged over different periods.
Table 2 – the use of the 2010-2020 period in this table is not really useful since it could easily be determined by adding together the other two columns, and is dominated so heavily by the latter period.
Line 194 – “overall” used 2x
Line 201 – Please check this calculation – 1493 citations of 82 articles is 18 citations per article, not 82.94 citations. This value (18), is not substantially higher than the UK 16.7. And the use of the term “significant” in line 202 implies some type of test, but I don’t believe any sort of statistical analysis was described.
Line 128 – comparing the 25 articles from 2010-2015 to the 1200+ from the latter period does not provide a balanced analysis of the topics, and the only real significant result would be to say that the number of publications has increased exponentially.
Line 131 – how were the keywords determined? Are these from the keywords associated with the article? From the title? This was not described clearly in the methods. Were the words determined prior to the analysis, or did they emerge from the analysis.
Line 134 – the use of the word “hotspot” implies a spatial component and I believe is not used correctly. This could be changed to something like “focal point”.
Line 136 – what is “risk removal”
Line 136 – this repeats the previous section and could be combined with the previous description to state what keywords were in common and what were new in the latter period.
Line 140 – the earlier literature focuses on where plastics are found, ie. The distribution and the characteristics. Not just the characteristic s.
Figure 2. Panel A of this figure and figure 3 are not informative because the trends are so dominated by the latter portion of the period and so the whole period trends are not distinguishable. Suggest removing these are presenting only the trends from the two 5 year periods.
Figure 2. Why is “high density polyethylene” used in A and C and just “polyethylene” used in B. there are a variety of other strange choices here. Including “polystyrene microplastic” but just “polypropylene” and “polyvinyl Chloride”. And many of the other words have “microplastic” combined with another term, which I don’t think is necessary and likely skews the analysis.
Line 216 – PP is also used a many types of single use packaging
Line 218 – the use of acronyms is not consistent
Line 225 – I’m not sure “methods” is the correct term here. Maybe type of plastic?
Line 230 – these studies don’t discuss much about degradation
Line 230 – this paper is from 2016, so past the cutoff for this period?
Line 250 – I agree that there were not a lot of studies of freshwater, but the majority of the studies called out here were done in freshwater.
Line 265 – this sentence is not clear
Line 272 – there are no “marine amphibians”. This study looks at amphipods. Again, proofreading by a native English speaker is important here
Line 280 – why HDPE here and PE previously. What about LDPE? Or studies that don’t specify? PE is pretty highly studied, but this splitting into different categories does not show this.
Line 281 – the last part of this sentence on ‘fibrous microplastics’ is not clear
Section 4.4 - is not on sources, really. The description is more focused on fate and adsorption rather than source. Suggest combining with the next section to avoid redundancy
Line 309 – this is not abundance this is concentration
Line 315 – I don’t follow what “obtained visually through microplastics in sewage” means?
Line 317 – I don’t believe that fugacity is the currect term here, but am not following the point of the sentence so cannot suggest an alternative.
Line 320 + - This summary of the literature seems to pick and choose a few sources but is not a comprehensive study, and the focus seems to be mostly on freshwater.
Line 347 – the study by Batel et al (2016) was a simple two species food chain from Artemia to zebrafish. Thus this description, which is repeated twice, of a food chain that goes from “shrimps to ungulates to zebrafish” is not only incorrectly citing the work, but also more or less impossible. Please check this reference.
Line 354 – the impact on humans has only very recently been evaluated. Most of the work to date looks only at the presence in the human body.
Line 356+ - this section immediately steps to complex interactions in the environment without addressing any inherent toxicity of the plastics themselves, or with plastic additives.
Line 370 – The differences between section 2) and section 3) are not clear – both seems to deal with adsorption of environmental contaminants onto plastics.
Line 385 – this doesn’t mesh with previous statement that older plastics have greater adsorption capacity
Line 386 – this section seems out of context and related to one specific antibiotic (hygromycin), but no study is cited in spite of the long description of this interaction.
Line 394- Not necessarily. Many studies still look at presence in organisms rather than effect.
Line 413 – this section is a lot of conjecture without citing any relevant studies
Line 421 – the impact of microplastic on genetics, inheritance, epigenetics, etc is a topic of intense study now and there are new publications coming out daily. A quick search for “microplastic” and “epigenetics” yielded many recent papers. Many newer studies look at gene transfer in biofilms, inherited traits associated with exposure, etc…
Line 466 – biogenetic is probably not the right term. Perhaps just genetic effects. But again, there are a lot of new studies coming out on this topic.
Reviewer 3 Report
Review toxics217242
General – This manuscript states its purpose is to document the status and trends of microplastics study based on different kinds of literature reviews. The manuscript is well written, but the section material needs to be adjusted for clarity and consistency.
The direction of the manuscript should be made consistent. It is not clear if the manuscript focus is documenting studies (initial statement), providing a technical review or giving subjective advice and comment. The initial text discusses the need to use more sophisticated review techniques for microplastics but the results do not make a comparison of these techniques. The introduction introduces microplastics research and not until the very end states aquatics will be the topic of concern. The discussion should have discussed the reason for the trends documented in the results, but instead often reads as a standard review of research or a subjective discussion. The conclusion should be a brief statement of what was found by the study, but reads more like a discussion and results section combined.
Some detailed comments are included below by line
43-44 what about animal and human transport
25 research on treatment and substitute or biodegradable products
58 feeding may not the best word here. consumption?
65 large sample is unclear
69 change introduction to need
77 awkward phrasing
78 if the goal is to look at aquatic issues then the preceding text should set that focus; the current text suggests this will be a general review
85 awkward phrasing
96-100 put in introduction
109 omit
115 growth rate not best wording
115-121 use the same statistics; average is matched with total
Fig 1 more detail needed in legend what were the sources and what was the statistic (total, average, aquatics) etc
125 why these number of papers
126 unclear
127 elaborate
130 single not best wording
134 do you mean hot topics?
139 wider research range unclear
145 are there statistics that could be used to quantify these conclusions
Table 1 these are only a few references; I suggest reducing the impacts to simple phrasing such as ingestion and including more authors
153-154 subjective conclusions not in keeping with results; just state quantitative findings in results
153 methods section should contain method for determining research and international cooperation (also in Fig 3 ) and linkages in other text
165 how did you quantify “spread more rapidly”
170 subjective and not appropriate for literature review statistics paper
191, 203 and throughout remove subjective statements
234 justify statement from results
244 this is not a paper on Microplastics but one on citations so stick to the number of citations in an area
253-254 unclear
293 decide if this is a paper on citation and research development or one on microplastics
342-344 omit – subjective
345-355 & section after 370 Here and elsewhere check the direction of the discussion. It is best to address the change in the number of papers that deal with food chain issues and the number that deal with different aspects of food chain issues over time. The discussion could then lead to the trends in research over time and discussion of why these may have occurred. The discussion sometimes reads like a standard review.
432 why was 2014-2016 a slow rise & explosive growth of what?
438-440 rephrase
Conclusion - The text reads more like a results section or personal opinion section; this should be a brief summary of findings for your study such as the number of papers on microplastics increased over time and the topics shifted from xxx to xxx etc.
Conclusion – future research is not a conclusion of your work and should be omitted. This could be a conclusion topic if it was presented in the results as a topic of discussion by researchers